# Technical note: A simple approach for efficient collection of field reference data for calibrating remote sensing mapping of northern wetlands

Magnus Gålfalk[1], Martin Karlson[1], Patrick Crill[2], Philippe Bousquet[3], David Bastviken[1]

[1]Department of Thematic Studies – Environmental Change, Linköping University, 581 83 Linköping, Sweden.
[2]Department of Geological Sciences, Stockholm University, 106 91 Stockholm, Sweden.
[3]Laboratoire des Sciences du Climat et de l'Environnement (LSCE), Gif sur Yvette, France.

*Correspondence to*: Magnus Gålfalk (magnus.galfalk@liu.se)

**Abstract.** The calibration and validation of remote sensing land cover products is highly dependent on accurate field reference data, which are costly and practically challenging to collect. We describe an optical method for collection of field reference data that is a fast, cost-efficient, and robust alternative to field surveys and UAV imaging. A light weight, water proof, remote controlled RGB-camera (GoPro HERO4 Silver, GoPro Inc.) was used to take wide-angle images from 3.1 - 4.5 m altitude using an extendable monopod, as well as representative near-ground (< 1 m) images to identify spectral and structural features that correspond to various land covers at present lighting conditions. A semi-automatic classification was made based on six surface types (graminoids, water, shrubs, dry moss, wet moss, and rock). The method enables collection of detailed field reference data which is critical in many remote sensing applications, such as satellite-based wetland mapping. The method uses common non-expensive equipment, does not require special skills or training, and is facilitated by a step-by-step manual that is included in the supplementary material. Over time a global ground cover database can be built that can be used as reference data for studies of non-forested wetlands from satellites such as Sentinel 1 and 2 (10 m pixel size).

## 1 Introduction

Accurate and timely land cover data are important for e.g. economic, political, and environmental assessments, and for societal and landscape planning and management. The capacity for generating land cover data products from remote sensing is developing rapidly. There has been an exponential increase in launches of new satellites with improved sensor capabilities, including shorter revisit time, larger area coverage, and increased spatial resolution (Belward & Skøien 2015). Similarly, the development of land cover products is increasingly supported by the progress in computing capacities and machine learning approaches.

At the same time it is clear that the knowledge of the Earth´s land cover is still poorly constrained. A comparison between multiple state-of-the-art land cover products for West Siberia revealed disturbing uncertainties (Frey and Smith 2007) as estimated wetland areas ranged from 2 - 26% of the total area, and the correspondence with *in situ* observations for wetlands

was only 2 - 56%. For lakes, all products revealed similar area cover (2-3%), but the agreement with field observations was as low as 0-5%. Hence, in spite of the progress in technical capabilities and data analysis progress, there are apparently fundamental factors that still need consideration to obtain accurate land cover information.

The West Siberia example is not unique. Current estimates of the global wetland area range from 8.6 to 26.9 x $10^6$ km$^2$ with great inconsistencies between different data products (Melton et al. 2013). The uncertainty in wetland distribution has multiple consequences, including being a major bottleneck for constraining the assessments of global methane ($CH_4$) emissions (Crill & Thornton 2017), which was the motivation for this area comparison. Wetlands and lakes are the largest natural $CH_4$ sources (Saunois et al. 2016) and available evidence suggest that these emissions can be highly climate sensitive, particularly at northern latitudes predicted to experience the highest temperature increases and melting permafrost – both contributing to higher $CH_4$ fluxes (Yvon-Durocher et al. 2014; Schuur et al. 2009).

$CH_4$ fluxes from areas with different plant communities in northern wetlands can differ by orders of magnitude (in the following, northern wetlands refers to non-forested boreal, subarctic, and arctic wetlands). Small wet areas dominated by emergent graminoid plants account for by far the highest fluxes per m$^2$, while the more widespread areas covered by e.g. Sphagnum mosses have much lower $CH_4$ emissions per m$^2$ (e.g. Bäckstrand et al. 2010). The fluxes associated with the heterogeneous and patchy (i.e. mixed) land cover in northern wetlands is well understood on the local plot scale, whereas the large-scale extrapolations are very uncertain. The two main reasons for this uncertainty is that the total wetland extent is unknown and that present map products do not distinguish between different wetland habitats which control fluxes and flux regulation. As a consequence the whole source attribution in the global $CH_4$ budget remains highly uncertain (Kirschke et al. 2013; Saunois et al. 2016).

Improved land cover products being relevant for $CH_4$ fluxes and their regulation are therefore needed to resolve this. The detailed characterization of wetland features or habitats requires the use of high resolution satellite data and sub-pixel classification that quantify percent, or fractional, land cover. A fundamental bottleneck for the development of fractional land cover products is the quantity and quality of the reference data used for calibration and validation (Foody 2013; Foody et al. 2016). In fact, reference data can often be any data available at higher resolution than the data product, including other satellite imagery, airborne surveys, in addition to field observations. In turn, the field observations can range from rapid landscape assessments to detailed vegetation mapping in inventory plots, where the latter yields high resolution and high-quality data but is very expensive to generate in terms of time and manpower (Olofsson et al. 2014; Frey & Smith 2007). Ground-based reference data for fractional land cover mapping can be acquired using traditional methods, such as visual estimation, point frame assessment or digital photography (Chen et al. 2010). These methods can be applied using a transect approach to increase the area coverage in order to match the spatial resolutions of different satellite sensors (Mougin et al. 2014).

The application of digital photography and image analysis software has shown promise for enabling rapid and objective measurements of fractional land cover that can be repeated over time for comparative analysis (Booth et al. 2006a). While several geometrical corrections and photometric setups are used, nadir (downward facing) and hemispherical view photography are most common, and the selected setup depends on the height structure of the vegetation (Chen et al. 2010).

Most previous research has however focused on distinguishing between major general categories, such as vegetation and non-vegetation (Laliberte et al. 2007; Zhou & Liu 2015), and are typically not used to characterize more subtle patterns within major land cover classes. Many applications in literature have been in rangeland, while there is a lack of wetland classification. Furthermore, images have mainly been close-up images taken from a nadir view perspective (Booth et al. 2006a; Chen et al. 2010; Zhou & Liu 2015), thereby limiting the spatial extent to well below the pixel size of satellite systems suitable for regional scale mapping.

From a methano-centric viewpoint, accurate reference data at high enough resolution, being able to separate wetland (and upland) habitats with differing flux levels and regulation, is needed to facilitate progress with available satellite sensors. The resolution should preferable be better than 1 $m^2$ given how the high emitting graminoid areas are scattered on the wettest spots where emergent plants can grow. Given this need, we propose a quick and simple type of field assessment adapted for the 10 x 10 m pixels of the Sentinel 1 and 2 satellites.

Our method uses true color images of the ground, followed by image analysis to distinguish fractional cover of key land cover types relevant for $CH_4$ fluxes from northern wetlands, where we focus on few classes, that differ in their $CH_4$ emissions. We provide a simple manual allowing anyone to take the photos needed in a few minutes per field plot. Land cover classification can then be made using the Red-Green-Blue (RGB) field images (sometimes also converting them to the Intensity-Hue-Saturation (IHS) color space) by software such as e.g. CAN-EYE (Weiss & Baret 2010), VegMeasure (Johnson et al. 2003), SamplePoint (Booth et al. 2006b), or eCognition (Trimble commercial software). With this simple approach it would be quick and easy for the community to share such images online and to generate a global reference database that can be used for land cover classification relevant to wetland $CH_4$ fluxes, of other purposes depending of the land cover classes used. We use our own routines written in Matlab due to the large field of view used in the method, in order to correct for the geometrical perspective when calculating areas (to speed up the development of a global land cover reference database, we can do the classification on request if all necessary parameters and images are available as given in our manual).

## 2 Field work

The camera setup is illustrated in Fig.1, with lines showing the spatial extent of a field plot. Our equipment included a lightweight RGB-camera (GoPro 4 Hero Silver; other types of cameras with remote control and suitable wide field of view would also work) mounted on an extendable monopod that allows imaging from a height of 3.1 - 4.5 m. The camera had a resolution of 4000 x 3000 pixels with a wide field of view (FOV) of 122.6 x 94.4 deg. and was remotely controlled over Bluetooth using a mobile phone application that allows a live preview, making it possible to always include the horizon close to the upper edge in each image (needed for image processing later – see below). The camera had a waterproof casing and could therefore be used in rainy conditions, making the method robust to variable weather conditions. Measurements were made for about 200 field plots in northern Sweden in the period 6-8 September 2016.

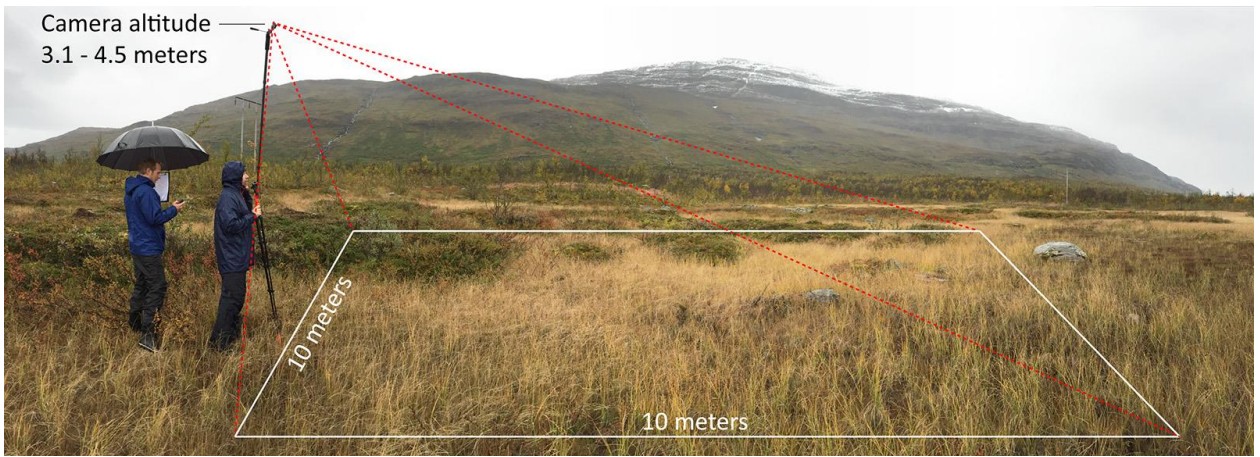

**Figure 1: A remotely controlled wide-field camera mounted on a long monopod captures the scene in one shot, from above the horizon down to nadir. After using the horizon image position to correct for the camera angle, a 10 x 10 m area close to the camera is used for classification.**

5    For each field plot, the following was recorded:

- One image taken at > 3.1 m height (see illustration in Fig. 1) which includes the horizon coordinate close to the top of the image.
- 3-4 close-up images of common surface cover in the plot (e.g. typical vegetation) and a very short note for each image indication what is shown, e.g. if a close-up image shows dry or wet moss (two of our classes) as there can be different

10       colors within a class.

- GPS position of the camera location (reference point)
- Notes of the image direction relative to the reference point.

A long modified monopod with a GoPro camera mounted at the end was used for the imaging. The geographic coordinate of the camera position was registered using a handheld Garmin Oregon 550 GPS with a horizontal accuracy of approximately 3

15    m. The positional accuracy of the images can be improved by using a differential GPS and by registering the cardinal direction of the FOV. The camera battery lasted for a few hours after a full charge, but was charged at intervals when not used, e.g. when moving between different field sites, making it possible to do all the imaging using only one camera.

## 3 Image processing and models

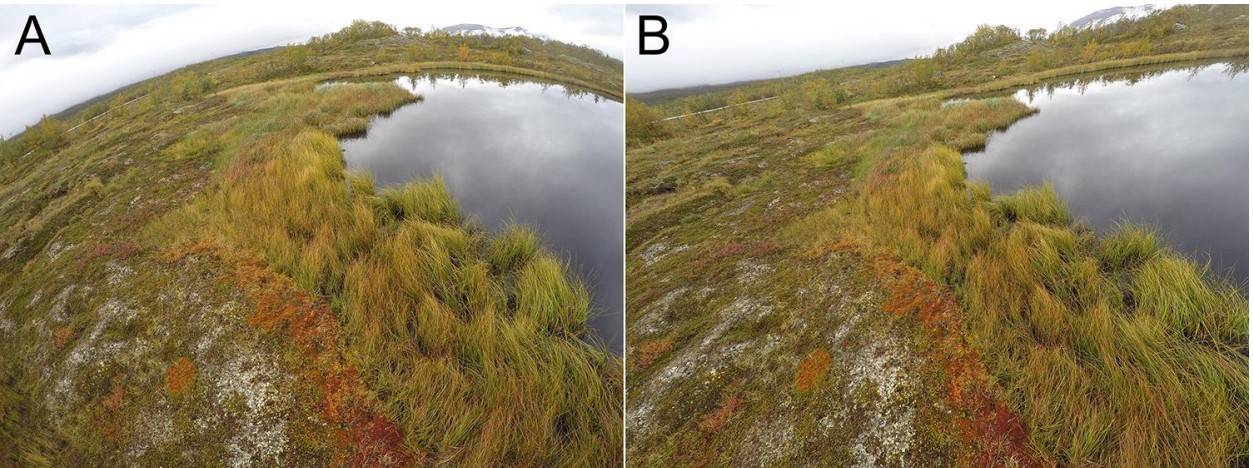

**Figure 2: Correction of lens distortion. (A) Raw wide-field camera image. (B) After correction.**

As the camera had a very wide FOV, the raw images do have a strong lens distortion (Fig. 2). This can be corrected for many

camera models (e.g., the GoPro series) using ready-made models in Adobe Lightroom or Photoshop, or by modelling the

distortion for any camera using the camera calibrator application in Matlab's computer vision system toolbox as described

below. A checkboard pattern is needed for the modelling (Fig. 3), which should consist of black and white squares with an

even number of squares in one direction and an odd number of squares in the other direction, preferably having a white boarder

around the pattern. The next step is to take images of this pattern from 10-20 unique camera positions, providing many

perspectives of the pattern with different angles for the distortion modelling (Fig. 3). In order to make accurate models it is

important both to have sharp images with no motion blur (e.g. due to movement or poor lighting) and to include several images

where the pattern is close to the edges of the image as this is where the distortion is greatest. An alternative but equivalent

method, preferably used for small calibration patterns printed on a standard sheet of paper (e.g. letter or A4), is to mount the

camera and instead move the paper to 10-20 unique positions having different angles to the camera. A quick way to do this

with only one person present is by recording a video while moving the paper slowly and then selecting calibration images from

the video afterwards. The illustration of camera positions used (Fig. 4) can also be displayed in the application as a mounted

camera with different positions of the calibration pattern. The next step is to enter the size of a checkerboard square (mm, cm,

or in) which is followed by an automatic identification of the corners of squares in the pattern (Fig. 3). Images with bad corner

detection can now be removed (optional) to improve the modelling. As a last step, camera parameters can now be calculated

by the press of a button and be saved as a variable in Matlab (cameraParams). This whole procedure only has to be done once

for each camera and field-of-view setting used, meaning once for a data collection campaign or a project if the same camera

model and field of view is used. Applying the correction to images is done using a single command in Matlab: img_corr =

undistortImage(img, cameraParams). Here img and img_corr are variables for the uncorrected and corrected versions,

respectively.

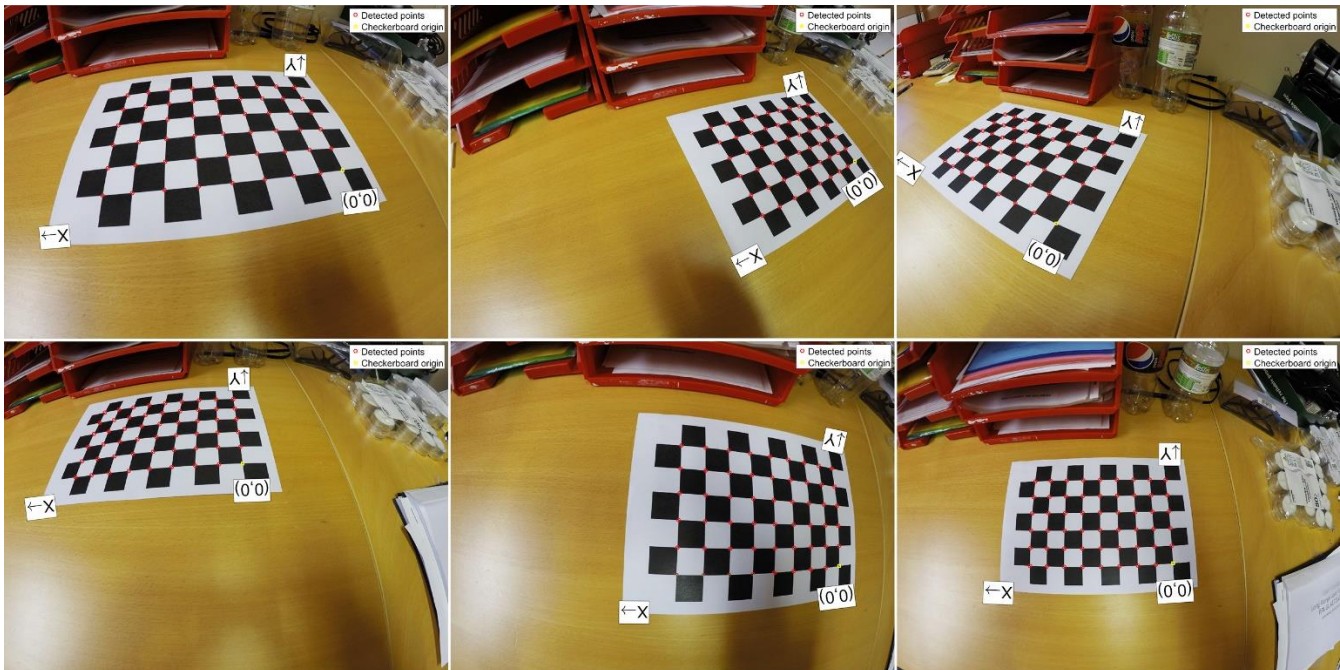

**Figure 3: Modelling of lens distortion. Checkboard pattern used for calibrations. Red circles are used to mark automatically detected square corners while the yellow square marks the origin of the coordinate system. Images of the pattern are taken from 10-20 different camera angles.**

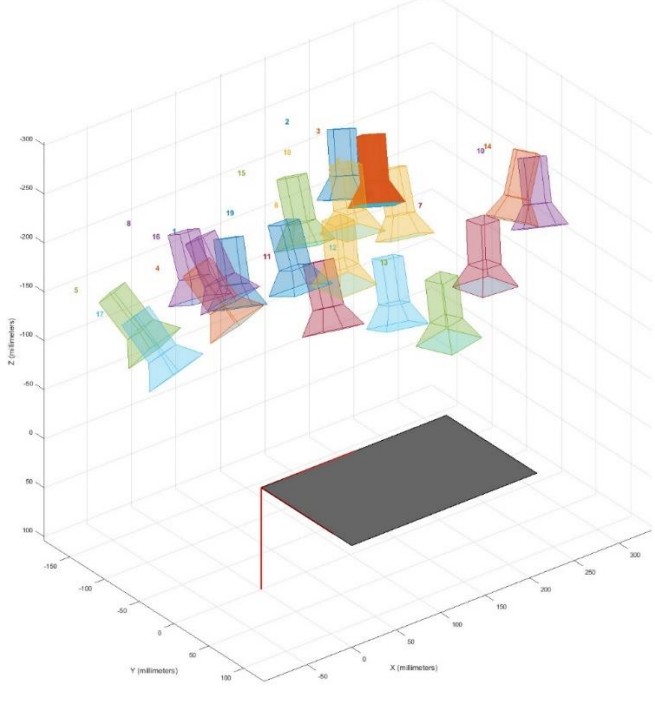

**Figure 4. Illustration using Matlab's Camera calibration application of the camera positions used when taking the calibration images.**

Using a distortion corrected calibration image, we then developed a model of the ground geometry by projecting and fitting a 10 x 10 m grid on a parking lot with measured distances marked using chalk (Fig. 5). Such calibration of projected ground

geometry only needs to be done when changing the camera model or field of view setting, and is valid for any camera height as long as the heights used in field and in the calibration imaging are known. It is done fast by drawing short lines every meter for distances up to 10 m, and some selected perpendicular lines at strategic positions to obtain the perspective. At least one, preferably two, perpendicular distances should be marked at a minimum of two different distances along the central line (in Fig. 5 at 2 and 4 m left of the central line at distances along the central line of 0 and 2.8 m). The geometric model uses the

camera FOV, camera height, and the vertical coordinate of the horizon (to obtain the camera angle). We find an excellent agreement between the modeled and measured grids (fits are within a few cm) for both camera heights of 3.1 and 4.5 m.

The vertical angle $\alpha$ from nadir to a certain point in the grid with ground distance $Y$ along the center line is given by $\alpha = \arctan(Y/h)$, where $h$ is the camera height. For distance points in our calibration image (Fig. 5), using 0.2 m steps in the range $0 - 1$ m and 1 m steps from 1 to 10 m, we calculate the nadir angles $\alpha(Y)$ and measure the corresponding vertical image

coordinates $y_{calib}(Y)$.

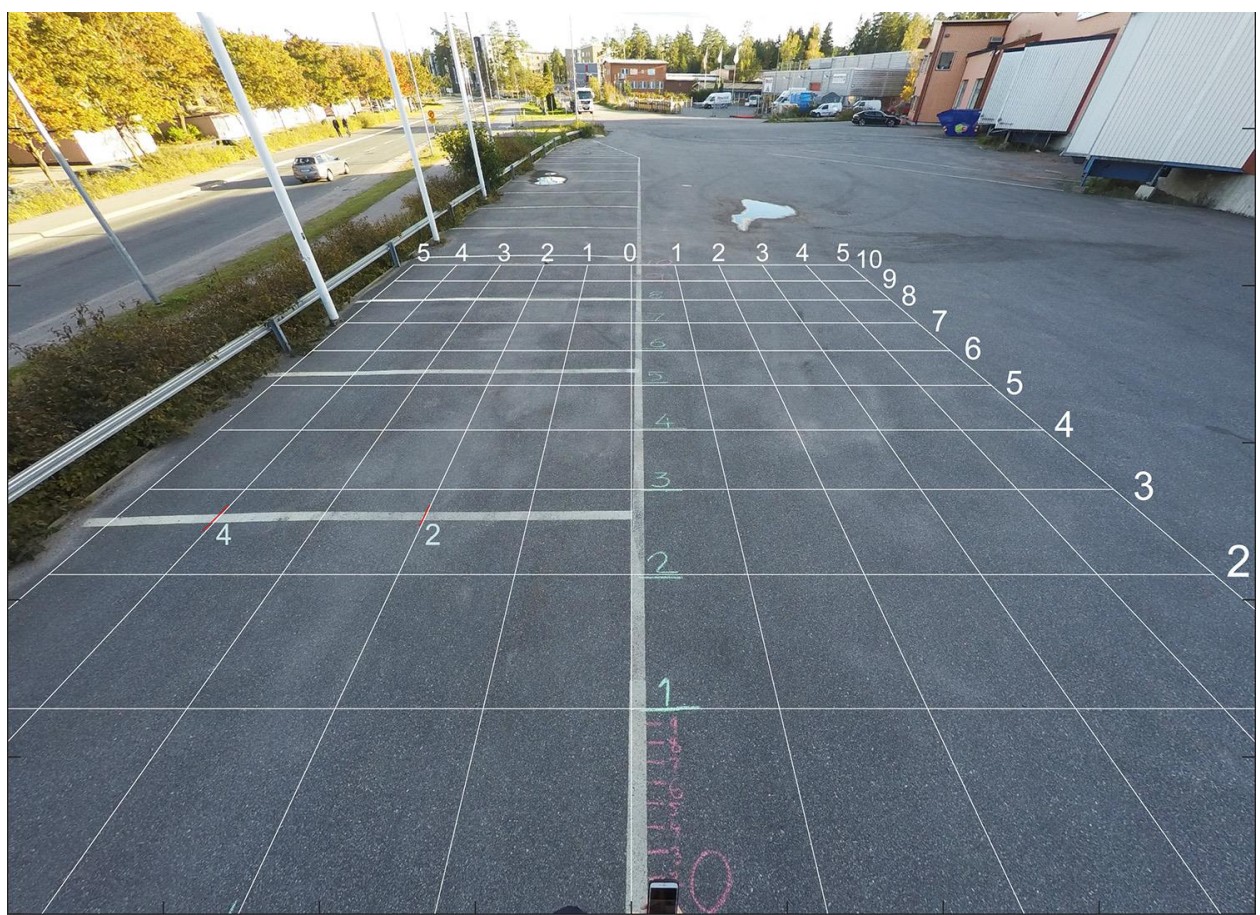

**Figure 5: Calibration of projected geometry using an image corrected for lens distortion. Model geometry are shown as white numbers and a white grid, while green and red numbers are written on the ground using chalk (red lines at 2 and 4 m left of the center line were strengthened for clarity). The camera height in this calibration measurement is 3.1 m.**

5      In principle, for any distortion corrected image there is a simple relationship $y_{img}(\alpha) = (\alpha(Y) - \alpha_0)/PFOV$, where $y_{img}$ is the image vertical pixel coordinate for a certain distance $Y$, $PFOV$ the pixel field of view (deg pixel$^{-1}$), and $\alpha_0$ the nadir angle of the bottom image edge. In practice, however, correction for lens distortion is not perfect so we have fitted a polynomial in the calibration image to obtain $y_{calib}(\alpha)$ from the known $\alpha$ and measured $y_{calib}$. Using this function we can then obtain the $y_{img}$ coordinate in any subsequent field image using

$$y_{img} = y_{calib}\left(\alpha + PFOV_{hor} \times \left(y_{img}^{hor} - y_{calib}^{hor}\right)\right) \tag{1}$$

10      where $y_{img}^{hor}$ and $y_{calib}^{hor}$ are the vertical image coordinates of the horizon in the field and calibration image, respectively. As the $PFOV$ varies by a small amount across the image due to small deviations in the lens distortion correction, we have used $PFOV_{hor}$ which is the pixel field of view at the horizon coordinate. In short, the shift in horizon position between the field and calibration images is used to compensate for the camera having different tilts in different images. In order to obtain correct ground geometry it is therefore important to always include the horizon in all images.

The horizontal ground scale $dx$ (pixels m⁻¹) varies linearly with $y_{img}$, making it possible to calculate the horizontal image coordinate $x_{img}$ using

$$x_{img} = x_c + X \times dx = x_c + X \times \left(y_{img}^{hor} - y_{img}\right) \times \frac{dx_0}{y_{calib}^{hor}} \times \frac{h_{calib}}{h_{img}} \tag{2}$$

where $dx_0$ is the horizontal ground scale at the bottom edge of the calibration image, $x_c$ the center line coordinate (half the horizontal image size), $X$ the horizontal ground distance, and $h_{calib}$ and $h_{img}$ the camera heights in the calibration and field image, respectively.

Thus, using Eqs. (1) and (2) we can calculate the image coordinates $(x_{img}, y_{img})$ in a field image from any ground coordinates $(X, Y)$. A model grid is shown in Fig. 5 together with the calibration image, illustrating their agreement.

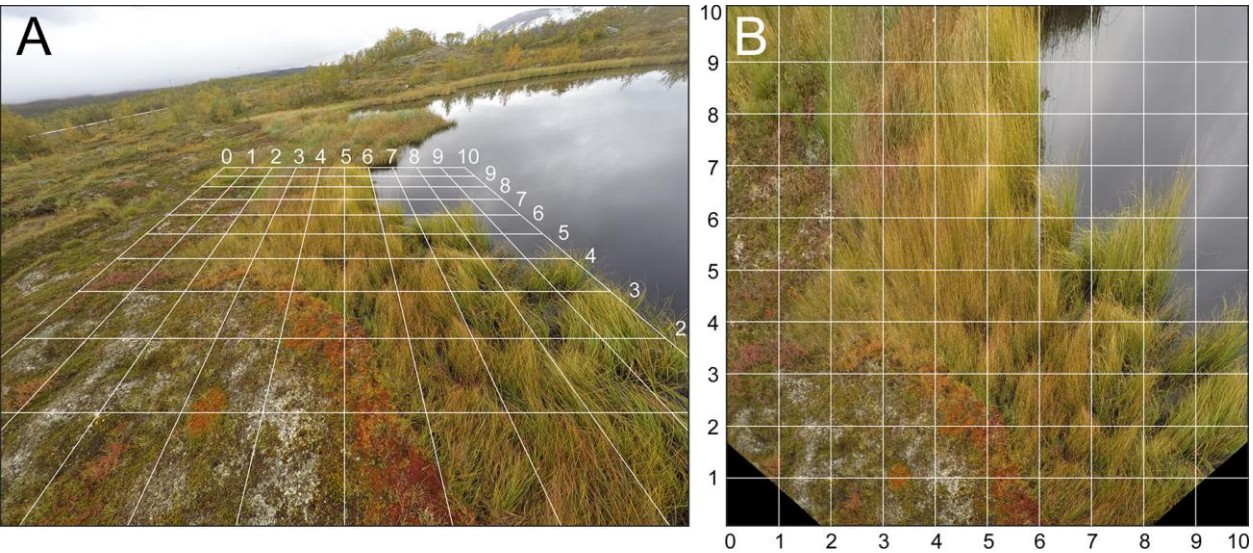

**Figure 6: One of our field plots. (A) Image corrected for lens distortion, with a projected 10 x 10 m grid overlaid. (B) Image after recalculation to overhead projection (10 x 10 m).**

For each field image, after correction for image distortion, our Matlab script asks for the $y$-coordinate of the horizon (which is selected using a mouse). This is used to calculate the camera tilt and to over-plot a distance grid projected on the ground (Fig. 6A). Using Eqs. (1) and (2) we then recalculate the image to an overhead projection of the nearest 10 x 10 m area (Fig. 6B). This is done using interpolation, where a $(x_{img}, y_{img})$ coordinate is obtained from each $(X, Y)$ coordinate, and the brightness in each color channel $(R, G, B)$ calculated using sub-pixel interpolation. The resulting image is reminiscent of an overhead image, with equal scales in both axes. There is however a small difference, as the geometry (due to line of sight) does not provide information about the ground behind high vegetation in the same way as an image taken from overhead. In cases with high vegetation (which is some of our 200 field plots), mostly high grass, we used a higher camera altitude to decrease obscured areas. Another possibility is to direct the camera towards nadir (see the manual in Supplementary material S1) to image areas -5 to +5 m from the center of a plot, further decreasing the viewing angles from nadir. We did not have any

problems with shrub or brushwood as it was only a couple of dm high, and Birch trees did not grow on the mires. We also recommend using a camera height of about 6 m to decrease obscuration and to increase the mapped area.

## 4 Image classification

After a field plot has been geometrically rectified, so that the spatial resolution is the same over the surface area used for classification, the script distinguishes land cover types by color, brightness and spatial variability. Aided by the close-up images of typical surface types also taken at each field plot (Fig. 7) and short field notes about the vegetation, providing further verification, a script is applied to each overhead-projected calibration field (Fig. 6B) that classifies the field plot into land cover types. This is a semi-automatic method that can account for illumination differences between images. In addition, it facilitates identification as there can for instance be different vegetation with similar color, and rock surfaces that have similar appearance as water or vegetation. After an initial automatic classification, the script has an interface that allows manual reclassification of areas between classes. The close-up images have high detail richness, allowing identification, and color and texture assignment of the different land cover classes during similar light and weather conditions as when the whole-plot image is taken. This makes results robust regarding different users collecting data, with respect to light conditions, times of day etc. The sensitivity is instead affected by the person defining the classes, just as with normal visual inspection.

For calculations of surface-color we filter the overhead projected field-images using a running 3 x 3 pixel mean filter, providing more reliable statistics. Spatial variation in brightness, used as a measurement of surface roughness, is calculated using a running 3 x 3 pixel standard deviation filter. Denoting the brightness in each (red, green, and blue) color channel $R$, $G$ and $B$, respectively, we could for instance find areas with green grass using the green filter index $2G/(R+B)$, where a value above 1 indicates green vegetation. In the same way, areas with water (if the close-up images show blue water due to clear sky) can be found using a blue filter index $2B/(R+G)$. If the close-up images show dark or gray water (cloudy weather) it can be distinguished from rock and white vegetation using either a total brightness index $(R+G+B)/3$ or an index that is sensitive to surface roughness, involving $\sigma(R)$, $\sigma(G)$, or $\sigma(B)$, where $\sigma$ denotes the 3 x 3 pixels standard deviation centered on each pixel, for a certain color channel.

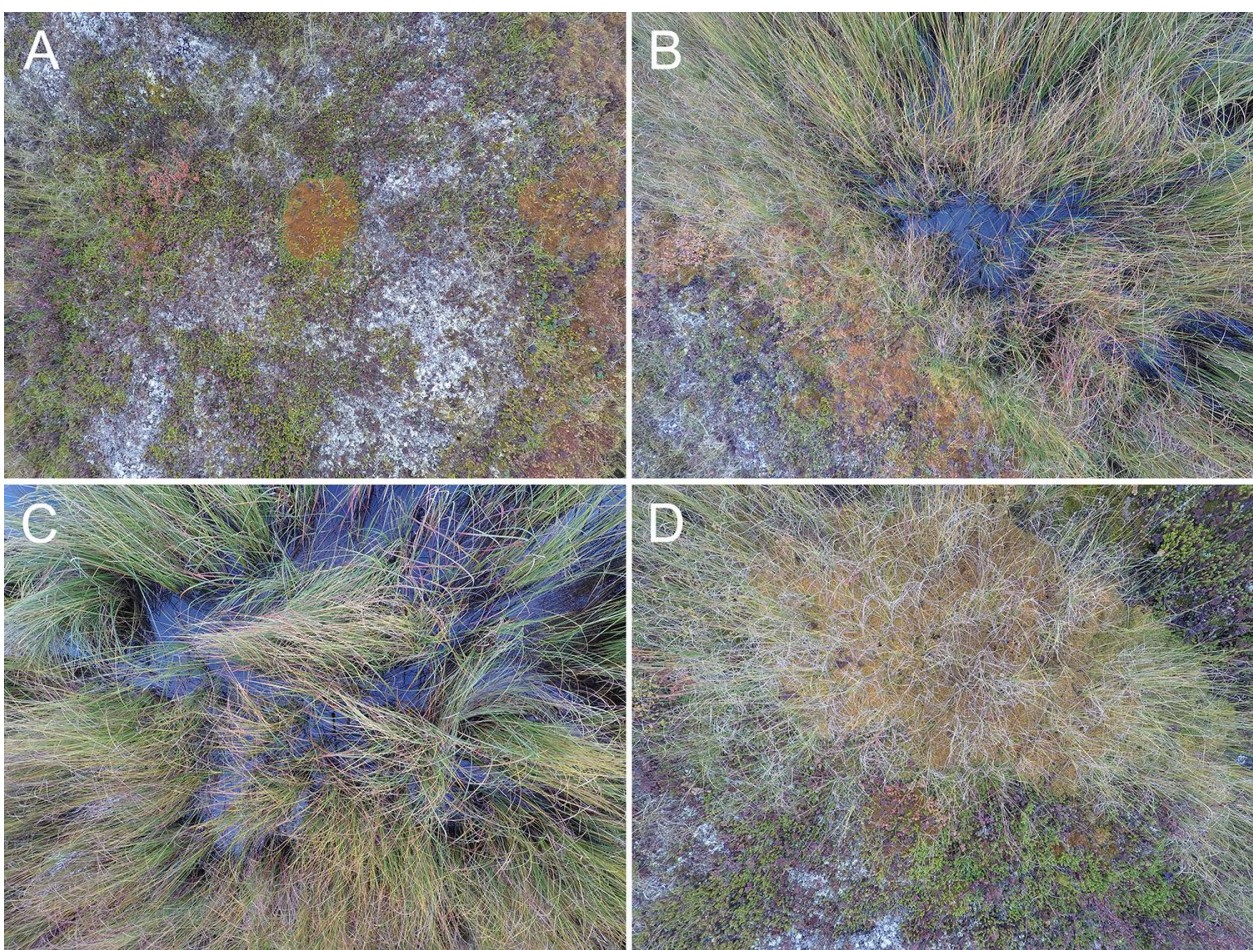

**Figure 7: Close-up images in one of our 10 x 10 m field plots (Fig. 6).**

In this study we used six different land cover types of relevance for $CH_4$ regulation: graminoids, water, shrubs, dry moss, wet moss, and rock. Examples of classified images are shown in Fig. 8. Additional field plots and classification examples can be found in supplementary material S2. Compared to the corrections for lens distortion and geometrical projection, the classification part often takes the longest time as it is semi-automatic and requires trial and error testing of which indices and class limits to use for each image as vegetation and lighting conditions might vary. After a number of images with similar vegetation and conditions have been classified the process goes much faster as the indices and limits will be roughly the same. It may also be needed to reclassify parts manually by moving a square region from one class to another based on visual inspection. The main advantage with this method of obtaining reference data is however that it is very fast in the field and works in all weather conditions. In a test study, we were able to make classifications of about 200 field plots in northern Sweden in a three-day test campaign despite rainy and windy conditions. For each field plot, surface area (m²) and coverage (%) were calculated for each class. The geometrical correction models (lens distortion and ground projection) was made in about an hour, while the classifications for all plots took a few days.

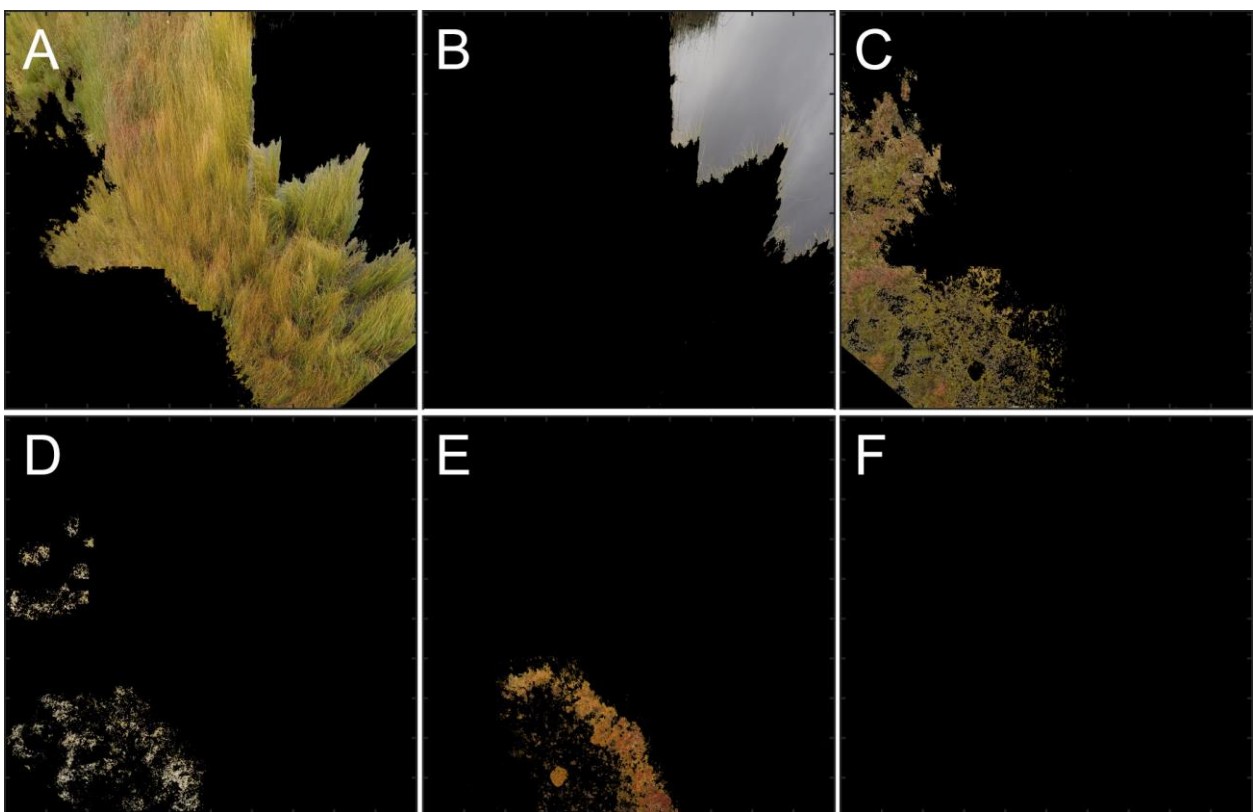

**Figure 8: Classification of a field plot image (Fig. 6B) into the six main surface components. All panels have an area of 10 x 10 m. (A) Graminoids. (B) Water. (C) Shrubs. (D) Dry moss. (E) Wet moss. (F) Rock.**

## 5 Conclusions

This study describes a quick method to document ground surface cover and process the data to make it suitable as reference data for remote sensing. The method requires a minimum of equipment that is frequently used by researchers and persons with general interest in outdoor activities, and image recording can be made easily and in a few minutes per plot without requirements of specific skills or training. In addition to covering large areas in a short time, it is a robust method that works in any weather using a waterproof camera. This provides an alternative to e.g. using small unmanned aerial vehicles (UAVs)

which are efficient at covering large areas, but have the drawbacks of being sensitive to both wind and rain and typically having flight times of about 20 minutes, considerably lower than this when many takeoffs and landings are needed when moving between plots. The presented photographic approach is also possible using a mobile phone camera, although such cameras usually have a very small field of view compared to many adventure cameras (such as the GoPro, which is also cheaper than a mobile phone). We recommend using a higher camera altitude; a height of 6 m would make mobile phone

imaging of 10 x 10 m possible (using a remote Bluetooth controller) and 20 x 20 m mapping using a camera with a large field of view such as the GoPro. Hence, if the method gets widespread and a fraction of those who visits northern wetlands (or other

environments without dense tall vegetation where the method is suitable) contributes images and related information, there is a potential for rapid development of a global database of images and processed results with detailed land cover for individual satellite pixels. In turn, this could become a valuable resource supplying reference data for remote sensing. To facilitate this development, supplementary material S1 includes a complete manual and authors will assist with early stage image processing
and initiate database development.

*Acknowledgements.* This study was funded by a grant from the Swedish Research Council VR to David Bastviken (ref. no. VR 2012-48). We would also like to acknowledge the collaboration with the IZOMET project (ref.no VR 2014-6584) and
IZOMET partner Marielle Saunois (Laboratoire des Sciences du Climat et de l'Environnement (LSCE), Gif sur Yvette, France).

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
