# Peer review of "Technical note: A simple approach for efficient collection of field reference data for calibrating remote sensing mapping of northern wetlands"

_Biogeosciences, 2017_

## Referee Comment (RC1) · Anonymous Referee #1 · 11 Nov 2017

This Tech Note presents a method for using a cheap robust consumer camera to capture RGB image data over (potentially) methane producing vegetation in wetlands, to aid validation of satellite land cover maps from 10 m pixel size. The aim is to provide a quick 'good enough' method to work over 10 m grids that can be collected and processed simply and rapidly.

The aim of the note is clear, it is well-written and provides a sound, simple approach, clearly explained. I think this could potentially provide a useful paper, as it is certainly true that collecting useful LC data at the plot to landscape scale is quite difficult, particularly in these kinds of areas where the vegetation is spatially variable, low stature

and with many areas of standing water, topography etc. However, in its current form the note is deficient in a couple of important aspects which I outline below. If these can be addressed in revision I think it could be published.

The main weakness is there is no quantitative assessment of the results, even at a very basic level. I realise the emphasis is on the collection and processing to the point where the user can choose their own method for generating LC info for comparison with other land cover products/maps i.e. the methods that might be applied for LC analysis are essentially down to the user after the processing of the RGB data. However, without some assessment of whether the resulting data are fit for purpose, we have no idea whether it's worth pursuing or not. At the very least I'd expect more than the 1 or 2 images presented, compared with some 'known' alternative assessment, just to give a first pass assessment of the approach. Another, related aspect, that's missing is some clear sensitivity assessment. How robust are results to different users collecting data, different light conditions, times of day and so on? You might even expect more than one camera to be looked at. Sure, GoPros are common and the methods *ought* to work for other cameras, but we don't know. Mobile phone camera data might potentially be even more useful. Also what about details of processing - various indices are mentioned but how sensitive are the results to these choices?

The manual part needs to have clear step by step guidance to the geom correction aspect as that is a key part of the method and may vary from camera to camera.

How long does it take to do the geometric calibration (which is one-off or only needs to be done occasionally) and the processing of an image to something useful? And how automatic is that latter process? The text implies some manual input is required. Clearly, again the onus is on the user and how they choose to do this, but the authors ought to give us an idea of timings for their workflow. If the aim is to do validation of LC data at anything other than trivial scales (few pixels), this is likely to require processing of 100s of images. If this requires significant manual input then the method may be limited. I note their generous offer to process data for users - but how prepared are

they to process a large amount of data from many users with many different systems? Is this feasible and for how long?

A more general issue that probably needs to be addressed at least in passing is cheap UAVs. The processing methods could be very similar (in the rgb sense), the geom is dealt with already by SfM software and you can cover much larger areas. Clearly this method is far more robust to wind, but is that the main / only benefit given the larger area UAVs can cover? My feeling is there is a place for this method *if* it is more clearly demonstrated, but it may be superseded very quickly (which is less likely if it can be applied to mobile phone data for example).

If the authors can clearly address these issues then I think a revised version would be suitable for publication.

---

## Referee Comment (RC2) · Anonymous Referee #2 · 16 Dec 2017

The issues this technical note seeks to remedy is truly an issue when working with spatial data and a hurdle for all researcher to overcome. The methods outlined in this research provide a low cost and easily accessible solution to not only researches but industry as well. Overall the paper is well written, and the methods seem useable. For a single researcher the methods provide a simple solution to a sometimes underestimated problem, especially those with limited time and resources. The even more important aspect to this research is that its simplicity and relatively inexpensive methods could lead to a database and data sharing. This is not possible with complex or expensive methods require resources and expert training and not to be underestimated. The equipment is both originally attainable on a budget but also common and

easily replaced if malfunctions or issues arise during a field campaign. The personnel needs are also cheaper and more available, making research and data collection more likely to occur. With a few edits outlined below this paper would be publishable and a help to benefit geospatial science as a whole. My first question is how this compares to the methods used currently. How long did it take to collect the 200 reference data points versus if they were collected in the "typical" fashion? (page 3) What is the computer processing time to create and classify the images and is that a major addition in time spent creating reference data? This system could potentially trade time in the field for time behind a computer time, not a bad option as processing time can be spread out over multiple analysts and a larger time frame. This methodology seems like it worked but did it? No discussion is made on how the results from the 200 plots performed and if they would help in classification. The level of accuracy achieved is an important metric for many and if this shows similar or a marginal decreases in accuracy with greatly reduced time or cost in data collection it would be well received. The risk of reduced accuracy may not be worth researcher changing the methods they currently use and some discussion on expected results of this new method should be included. The clarification of this topic would explain how data is integrated into classification of land cover and how this method compares to the current methods in terms of time savings and accuracy estimates? The section on distortion models is important for the research and could throw off all results for anyone using it. (page 4) What model was used and why was it used, enough explanation should be included for researchers to understand the difference distortion models would make on results and the sensitivity of different models. This is important especially if comparisons between research groups or sharing of data is going to occur. Calling validation "ground truth" should not be done, even if it's explained. The fact that it was explained means that there is an understanding that it's a bad term, call it reference data and remove any reference to "ground truth." The next sentence goes on to state 100% accurate reference data is not possible, reference data should be 100% accurate within the margin of possible error, it is not however 100% guaranteed that it represents the population or a large

study area, nor is it 100% guaranteed due to geolocation error that it is correctly located in your image compared to the true ground location. Clarification on this topic by the author would be helpful. Higher vegetation causing problems is made (page 7), but what is high vegetation and when does it start to deteriorate the data? Was high vegetation seen in the 200 plots created here and at what rate did tall vegetation cause problems? Knowing this would allow a decision on the applicability of this method to different research. My final comments on this technical note are about the writing mechanics. The sentences and paragraphs are well written, however in a few cases they start weak with; however, for example, to resolve this, etc. An effort should be made to start sentences with the primary subject of the sentence and tighten up some of the language and remove extra words seen throughout the paper. The acknowledgement heading is floating on page 10 line 15 as well. The additions and clarifications outlined above would make this an article worth publishing.

---

## Author Comment (AC3) · 15 Jan 2018

1. Quantitative assessment. We agree with the point that more classifications could be included in the manuscript to illustrate that the method delineates the land cover types accurately. However, we cannot see how we could get hold of reference material to validate our plots, because the results are based on information very similar to what is experienced if being on site doing a manual land cover inventory. The validation is done by the person who takes the close-up images that represent the different classes (acting as reference data as the view of these close-up images has the same detail richness as if standing on-site in situ), and then the person that does the image

processing when setting thresholds and manually cleaning up the classification. The accuracy is therefore related to the classification interpretation of the person taking the close-up images and the image processer. To make it more robust, several interpreters can be used. In this case, three of the authors (Gålfalk, Karlsson and Bastviken) visited all sites and discussed the image interpretation, so the classification is based on their joint assessment at each site. In the end, this approach is similar to in-situ visual inspection of surface cover types, which are regarded as fundamental reference data, although subjective and dependent on the knowledge of the person(s) doing the inspections. The difference is that our suggested method is much faster in the field. - We have now added three classification examples in Supplementary material S2, which means that there are now five examples of classifications in the manuscript (one in the main text and four in S2).

2. Sensitivity assessment. Any camera can be used, and we have now clarified this by adding a custom step-by-step guide of how the image distortion is done for a custom camera in Matlab using the built in Camera calibration application. The problem with mobile phones is that the field of view is often really narrow, requiring a much higher viewing altitude in the field to map 10 x 10 m which could become impractical and less useful. The method is however just as valid for any other cameras than a GoPro; we just used a GoPro for the very large field of view, and because it is a very popular cheap light-weight camera that is easily available with many possibilities for mounting as it has lots of accessories for this. Doubling the camera altitude (to about 6 meters) 10 x 10 meters could be mapped with a mobile phone, which could be controlled by a bluetooth remote control, and 20 x 20 meters mapped with a GoPro which would be a good area for e.g. 10 x 10 meters Sentinel satellite pixels. About the indices and sensitivity for different choices – the method provides a palette of potential discriminators (colors, indices and textures) from which the user needs to identify the most appropriate ones based on trial and error. It will depend on the lighting conditions (e.g. water reflecting clouds or a blue sky), the white balance of the camera, and the color of different vegetation. Importantly, as the close-up images have high detail richness, they

allow identification, and color and texture assignment of the different land cover classes during similar light and weather conditions as when the whole-plot image is taken. - This has now been further clarified in the manuscript (image classification section) and makes results robust regarding different users collecting data, with respect to light conditions, times of day etc. The sensitivity is instead affected by the person defining the classes, just as with normal visual inspection. - We have also added in the manuscript that it is important to write down notes for each close-up image, making a judgement in the field of the vegetation in each of the three close-up images per plot. This will aid in the calibration process as e.g. a class can contain several vegetation types, having different colors, that are classified separately and merged into the same class. - We have added to the manuscript that we recommend a higher camera altitude, and that it would be possibility to use a mobile phone. - We have added a custom step-by-step guide of how the image distortion is done for a custom camera in Matlab.

3. Geometric correction. The time taken to classify the images depend on the user, and is a trial-and-error process, involving trying different color differences (such as Green – Red) and setting limits of the classes. The manual part, setting these limits and reclassifying parts of the image, could be a very small effort for some images but take more time for others, and is often the step that takes the most time – compared to correction of lens distortion and ground geometry. However, as images involving different vegetation (say about 10 plots) have been classified, the classification becomes much faster as there will be similar vegetation and thus similar color differences for separating vegetation. - A clear step-by-step guide has now been added showing how to do a geometric lens distortion correction for any camera. We have also added that this only has to be done once, when using a new camera or a new field of view setting. This means once for a whole data collection campaign or a whole project (if the same camera model and field of view is used). - We have now added information about the time used for different steps in our workflow in section 4: "In a test study, we were able to make classifications of about 200 field plots in northern Sweden in a three-day test campaign despite rainy and windy conditions. For each field plot, surface area (m2)

and coverage (%) were calculated for each class. The geometrical correction models (lens distortion and ground projection) was made in about an hour, while the classifications for all plots took a few days." - The same applies to the calibration of the projected geometry, it only has to be done once for a certain camera and field of view setting. We have added a description to the manual in Supplementary material S1 about this. - The geometric correction process is fast and fully automatic (only the height of the camera needs to be entered). We have also added a clearer description of how the calibration images should be obtained for the geometrical projection.

4. General issues – UAVs and mobile phones. UAVs can be more advantageous if only one larger area is to be mapped in greater detail per day and if it is possible to wait for favorable weather conditions. However, if there is a need to visit many locations in a region without time to wait for good weather, UAVs (especially cheap ones below 10 000 USD) have several disadvantages. They can only fly at low to medium wind speeds and cannot fly when it is raining. Flight times are often around 20 minutes (when continuously in the air), but become much shorter when many takeoffs and landings are made, which would be the case when moving between plots (in our case about 70 plots per day) and can only take off or land on flat dry surfaces, unless hand-cached which is a risk in itself for larger more robust UAVs. A very large amount of batteries would have to be carried as they take too long time to recharge during the measurements. As for mobile phone cameras, yes, they can be used with our method, but field of views are often so small that this becomes very inefficient. Mobile phones could work efficiently if a wide-angle lens was mounted to the mobile phone but then it just becomes a less robust and more expensive adventure camera (also having similar image distortion from the added lens). A dedicated adventure camera (e.g. GoPro) is cheaper than a mobile phone or a drone, which is advantageous when funding is limited or for citizen science efforts. - We have added a paragraph about this (using UAVs or mobile phones) in the conclusions. "Small Unmanned aerial vehicles (UAVs) have also been considered as they are very efficient at covering large areas, they however have the drawbacks of being sensitive to both wind and rain and typically having flight times

of about 20 minutes, considerably lower than this when many takeoffs and landings are needed when moving between plots. Another option is using a mobile phone camera, this would work using our proposed method, but would have a very small field of view compared to many adventure cameras (such as the GoPro, which is also cheaper than a mobile phone). We recommend using a higher camera altitude; a height of 6 meters would make mobile phone imaging of 10 x 10 meters possible (using a remote bluetooth controller) and 20 x 20 meters mapping using a camera with a large field of view such as the GoPro."

We would like to thank both anonymous reviewers for their valuable comments for improving the manuscript.

A

B

C  D  E

**Fig. 1.**

[Figure]

**Fig. 2.**

**Fig. 3.**

[Figure]

**Fig. 4.**

[Figure]

**Fig. 5.**

---

## Author Comment (AC4) · 15 Jan 2018

1. Comparison to other (similar) methods. It is true that less time in the field is traded in for more time behind a computer classifying images semi-automatically (compared to the traditional methods). We agree that this is a nice tradeoff, and in addition that the method is very robust for different weather conditions (high wind, rain) which would not be the case for a e.g. a drone (see answer to RC #1–4). Traditional methods are described on page 2 (visual estimation, point frame assessment and digital photography), and are generally slow on large areas such as 10 x 10 m (or even 20 x 20 m which is possible using a camera height of 6 meters which is now suggested in the manuscript)

as they require individual attention to multiple smaller sub-areas e.g. 1 x 1 m represent-ing grid cells of the larger area. If extent of individual land cover classes are measured accurately in each such grid cell the traditional methods are very slow. Therefore, land cover fractions in each grid is often arbitrarily estimated visually to speed up assess-ments, likely resulting in lower accuracy and greater subjectivity among persons, than photographic methods such as proposed here. Digital photography is faster, but often downward facing cameras from small heights have been used. Our method captures 10 x 10 m in one image (seconds in the field) and then a few close-up images for reference of typical land cover under current light and weather conditions. - We have now added information about the time used for different steps in our workflow in sec-tion 4: "In a test study, we were able to make classifications of about 200 field plots in northern Sweden in a three-day test campaign despite rainy and windy conditions. For each field plot, surface area (m2) and coverage (%) were calculated for each class. The geometrical correction models (lens distortion and ground projection) was made in about an hour, while the classifications for all plots took a few days."

2. Accuracy assessment. See answers to RC #1–1 and RC #2–1.

3. Geometric correction. See answer to RC #1–3.

4. Use of terms. We have now changed the term "ground truth" to "reference data" throughout the text.

5. Limitations for high vegetation. High vegetation is for example high grass type vegetation that is also dense enough to obscure the ground behind it. Increasing the camera height will decrease this potential problem, and it will be worse for large dis-tances (near the edge of the field of view) as the viewing angles increases from nadir. For short grass, rocks etc. we did not have any problems from this, neither did we have problems from Birch trees as they do not grow on the mires and the shrub/brushwood was only a couple of decimeters high. For plots with high vegetation we used a larger camera height for this reason. Another solution for such plots could be to direct the

camera towards nadir, making angles smaller and the obscuration less – this is mentioned in the supplementary information S1 (manual), step 2, "Alternative 2 is to stand in the center of the plot and. . . as tall vegetation will not obscure the view towards lower vegetation as much". - We have now added a paragraph on vegetation height, our experiences, and solutions, at the end of section 3: "There is however a small difference, as the geometry (due to line of sight) does not provide information about the ground behind high vegetation in the same way as an image taken from overhead. In cases with high vegetation (which is some of our 200 field plots), mostly high grass, we used a higher camera altitude to decrease obscured areas. Another possibility is to direct the camera towards nadir (see the manual in Supplementary material S1) to image areas -5 to +5 meters from the center of a plot, further decreasing the viewing angles from nadir. We did not have any problems with shrub or brushwood as it was only a couple of decimeters high, and Birch trees did not grow on the mires. We also recommend using a camera height of about 6 meters to decrease obscuration and to increase the mapped area."

6. Writing. The acknowledgement heading has now been moved closer to the acknowledgement text. The start of sentences have also been improved throughout the paper.

We would like to thank both anonymous reviewers for their valuable comments for improving the manuscript.

---

## Author Response (AR1)

**Revision Biogeosciences**

**RC #1**

**1. Quantitative assessment**

The main weakness is there is no quantitative assessment of the results, even at a very basic level. Without some assessment of whether the resulting data are fit for purpose, we have no idea whether it's worth pursuing or not. At the very least I'd expect more than the 1 or 2 images presented, compared with some 'known' alternative assessment, just to give a first pass assessment of the approach.

**Answer:** We agree with the point that more classifications could be included in the manuscript to illustrate that the method delineates the land cover types accurately. However, we cannot see how we could get hold of reference material to validate our plots, because the results are based on information very similar to what is experienced if being on site doing a manual land cover inventory. The validation is done by the person who takes the close-up images that represent the different classes (acting as reference data as the view of these close-up images has the same detail richness as if standing on-site in situ), and then the person that does the image processing when setting thresholds and manually cleaning up the classification. The accuracy is therefore related to the classification interpretation of the person taking the close-up images and the image processer. To make it more robust, several interpreters can be used. In this case, three of the authors (Gålfalk, Karlsson and Bastviken) visited all sites and discussed the image interpretation, so the classification is based on their joint assessment at each site. In the end, this approach is similar to in-situ visual inspection of surface cover types, which are regarded as fundamental reference data, although subjective and dependent on the knowledge of the person(s) doing the inspections. The difference is that our suggested method is much faster in the field.

Changes made:

- We have now added three classification examples in Supplementary material S2 (Figures S4, S5, and S6 on pages 20-22), which means that there are now five examples of classifications in the manuscript (one in the main text and four in S2).

**2. Sensitivity assessment**

How robust are results to different users collecting data, different light conditions, times of day and so on?

You might even expect more than one camera to be looked at. Sure, GoPros are common and the methods *ought* to work for other cameras, but we don't know. Mobile phone camera data might potentially be even more useful.

Also what about details of processing - various indices are mentioned but how sensitive are the results to these choices?

**Answer:** Any camera can be used, and we have now clarified this by adding a custom step-by-step guide of how the image distortion is done for a custom camera in Matlab using the built in Camera calibration application. The problem with mobile phones is that the field of view is often really narrow, requiring a much higher viewing altitude in the field to map 10 x 10 m which could become impractical and less useful. The method is however just as valid

for any other cameras than a GoPro; we just used a GoPro for the very large field of view, and because it is a very popular cheap light-weight camera that is easily available with many possibilities for mounting as it has lots of accessories for this. Doubling the camera altitude (to about 6 meters) 10 x 10 meters could be mapped with a mobile phone, which could be controlled by a Bluetooth remote control, and 20 x 20 meters mapped with a GoPro which would be a good area for e.g. 10 x 10 meters Sentinel satellite pixels.

About the indices and sensitivity for different choices – the method provides a palette of potential discriminators (colors, indices and textures) from which the user needs to identify the most appropriate ones based on trial and error. It will depend on the lighting conditions (e.g. water reflecting clouds or a blue sky), the white balance of the camera, and the color of different vegetation.

Importantly, as the close-up images have high detail richness, they allow identification, and color and texture assignment of the different land cover classes during similar light and weather conditions as when the whole-plot image is taken.

Changes made:

- This has now been further clarified in the manuscript (page 10, lines 12-15) and makes results robust regarding different users collecting data, with respect to light conditions, times of day etc. The sensitivity is instead affected by the person defining the classes, just as with normal visual inspection.

- We have also added in the manuscript that it is important to write down notes for each close-up image, making a judgement in the field of the vegetation in each of the three close-up images per plot (page 10, line 7; page 16, line 3-4). This will aid in the calibration process as e.g. a class can contain several vegetation types, having different colors, that are classified separately and merged into the same class.

- We have added to the manuscript that we recommend a higher camera altitude (page 10, lines 2-3; page 12, lines 14-15; page 15, line 6), and that it would be possibility to use a mobile phone (page 12, lines 14-16)

- We have added a custom step-by-step guide of how the image distortion is done for any custom camera in Matlab (pages 5-6).

**3. Geometric correction**

The manual part needs to have clear step by step guidance to the geometric correction aspect as that is a key part of the method and may vary from camera to camera.

How long does it take to do the geometric calibration (which is one-off or only needs to be done occasionally) and the processing of an image to something useful? How automatic is the geometric correction process? The text implies some manual input is required. Clearly, again the onus is on the user and how they choose to do this, but the authors ought to give us an idea of timings for their workflow. If the aim is to do validation of LC data at anything other than trivial scales (few pixels), this is likely to require processing of 100s of images. If this requires significant manual input then the method may be limited. I note their generous offer to process data for users - but how prepared are they to process a large amount of data from many users with many different systems? Is this feasible and for how long?

**Answer:**

The time taken to classify the images depend on the user, and is a trial-and-error process, involving trying different color differences (such as Green – Red) and setting limits of the classes. The manual part, setting these limits and reclassifying parts of the image, could be a very small effort for some images but take more time for others, and is often the step that takes the most time – compared to correction of lens distortion and ground geometry. However, as images involving different vegetation (say about 10 plots) have been classified, the classification becomes much faster as there will be similar vegetation and thus similar color differences for separating vegetation.

Changes made:

- A clear step-by-step guide has now been added showing how to do a geometric lens distortion correction for any camera (pages 5-6). We have also added that this only has to be done once, when using a new camera or a new field of view setting (page 7, lines 4-9). This means once for a whole data collection campaign or a whole project (if the same camera model and field of view is used).

- We have now added information about the time used for different steps in our workflow in section 4: "In a test study, we were able to make classifications of about 200 field plots in northern Sweden in a three-day test campaign despite rainy and windy conditions. For each field plot, surface area (m2) and coverage (%) were calculated for each class. The geometrical correction models (lens distortion and ground projection) was made in about an hour, while the classifications for all plots took a few days." (page 11, lines 11-14)

- The same applies to the calibration of the projected geometry, it only has to be done once for a certain camera and field of view setting. (page 7, lines 4-6)

- The geometric correction process is fast and fully automatic (only the height of the camera needs to be entered). We have also added a clearer description of how the calibration images should be obtained for the geometrical projection. (page 7, lines 5-9).

**4. General issues.**

A more general issue that probably needs to be addressed at least in passing is cheap UAVs. The processing methods could be very similar (in the rgb sense), the geom is dealt with already by SfM (Structure from Motion?) software and you can cover much larger areas. Clearly this method is far more robust to wind, but is that the main / only benefit given the larger area UAVs can cover? My feeling is there is a place for this method *if* it is more clearly demonstrated, but it may be superseded very quickly (which is less likely if it can be applied to mobile phone data for example).

**Answer:** UAVs can be more advantageous if only one larger area is to be mapped in greater detail per day and if it is possible to wait for favorable weather conditions. However, if there is a need to visit many locations in a region without time to wait for good weather, UAVs (especially cheap ones below 10 000 USD) have several disadvantages. They can only fly at low to medium wind speeds and cannot fly when it is raining. Flight times are often around 20 minutes (when continuously in the air), but become much shorter when many takeoffs and landings are made, which would be the case when moving between plots (in our case about 70 plots per day) and can only take off or land on flat dry surfaces, unless hand-cached which

is a risk in itself for larger more robust UAVs. A very large amount of batteries would have to be carried as they take too long time to recharge during the measurements.
As for mobile phone cameras, yes, they can be used with our method, but field of views are often so small that this becomes very inefficient.

Mobile phones could work efficiently if a wide-angle lens was mounted to the mobile phone but then it just becomes a less robust and more expensive adventure camera (also having similar image distortion from the added lens). A dedicated adventure camera (e.g. GoPro) is cheaper than a mobile phone or a drone, which is advantageous when funding is limited or for citizen science efforts.

Changes made:

- We have now added a paragraph about using UAVs or mobile phones in the conclusions. (page 12, lines 9-16)

We would like to thank both anonymous reviewers for their valuable comments for improving the manuscript.

**RC #2**

**1. Comparison to other (similar) methods** (My first question is how this compares to the methods used currently)

How long did it take to collect the 200 reference data points versus if they were collected in the "typical" fashion? (page 3)

What is the computer processing time to create and classify the images and is that a major addition in time spent creating reference data? This system could potentially trade time in the field for time behind a computer time, not a bad option as processing time can be spread out over multiple analysts and a larger time frame.

**Answer:**

It is true that less time in the field is traded in for more time behind a computer classifying images semi-automatically (compared to the traditional methods). We agree that this is a nice tradeoff, and in addition that the method is very robust for different weather conditions (high wind, rain) which would not be the case for a e.g. a drone (see RC #1–4).

Traditional methods are described on page 2 (visual estimation, point frame assessment and digital photography), and are generally slow on large areas such as 10 x 10 m (or even 20 x 20 m which is possible using a camera height of 6 meters which is now suggested in the manuscript) as they require individual attention to multiple smaller sub-areas e.g. 1 x 1 m representing grid cells of the larger area. If extent of individual land cover classes are measured accurately in each such grid cell the traditional methods are very slow. Therefore, land cover fractions in each grid is often arbitrarily estimated visually to speed up assessments, likely resulting in lower accuracy and greater subjectivity among persons, than photographic methods such as proposed here. Digital photography is faster, but often downward facing cameras from small heights have been used. Our method captures 10 x 10

m in one image (seconds in the field) and then a few close-up images for reference of typical land cover under current light and weather conditions.

Changes made:

- We have now added information about the time used for different steps in our workflow in section 4: "In a test study, we were able to make classifications of about 200 field plots in northern Sweden in a three-day test campaign despite rainy and windy conditions. For each field plot, surface area (m2) and coverage (%) were calculated for each class. The geometrical correction models (lens distortion and ground projection) was made in about an hour, while the classifications for all plots took a few days.". (page 11, lines 11-14).

**2. Accuracy assessment** (This methodology seems like it worked but did it?)

No discussion is made on how the results from the 200 plots performed and if they would help in classification. The level of accuracy achieved is an important metric for many and if this shows similar or a marginal decreases in accuracy with greatly reduced time or cost in data collection it would be well received. The risk of reduced accuracy may not be worth researcher changing the methods they currently use and some discussion on expected results of this new method should be included. The clarification of this topic would explain how data is integrated into classification of land cover and how this method compares to the current methods in terms of time savings and accuracy estimates?

**Answer:** See RC #1–1 and RC #2–1

**3. Geometric correction** (The section on distortion models is important for the research and could throw off all results for anyone using it. (page 4)).

What model was used and why was it used, enough explanation should be included for researchers to understand the difference distortion models would make on results and the sensitivity of different models. This is important especially if comparisons between research groups or sharing of data is going to occur.

**Answer:** See RC #1–3

**4. Use of terms** (ground truth data)
Calling validation "ground truth" should not be done, even if it's explained. The fact that it was explained means that there is an understanding that it's a bad term, call it reference data and remove any reference to "ground truth." The next sentence goes on to state 100% accurate reference data is not possible, reference data should be 100% accurate within the margin of possible error, it is not however 100% guaranteed that it represents the population or a large study area, nor is it 100% guaranteed due to geolocation error that it is correctly located in your image compared to the true ground location. Clarification on this topic by the author would be helpful.

**Answer:**

Changes made:

We have now changed the term "ground truth" to "reference data" throughout the text.

**5. Limitations for high vegetation (Higher vegetation causing problems is made (page 7))**

What is high vegetation and when does it start to deteriorate the data? Was high vegetation seen in the 200 plots created here and at what rate did tall vegetation cause problems? Knowing this would allow a decision on the applicability of this method to different research.

**Answer:** High vegetation is for example high grass type vegetation that is also dense enough to obscure the ground behind it. Increasing the camera height will decrease this potential problem, and it will be worse for large distances (near the edge of the field of view) as the viewing angles increases from nadir. For short grass, rocks etc. we did not have any problems from this, neither did we have problems from Birch trees as they do not grow on the mires and the shrub/brushwood was only a couple of decimeters high.

For plots with high vegetation we used a larger camera height for this reason. Another solution for such plots could be to direct the camera towards nadir, making angles smaller and the obscuration less – this is mentioned in the supplementary information S1 (manual), step 2, "Alternative 2 is to stand in the center of the plot and… as tall vegetation will not obscure the view towards lower vegetation as much".

Changes made:

- We have now added a paragraph on vegetation height, our experiences, and solutions, at the end of section 3 (page 9 line 16 – page 10 line 3):

"There is however a small difference, as the geometry (due to line of sight) does not provide information about the ground behind high vegetation in the same way as an image taken from overhead. In cases with high vegetation (which is some of our 200 field plots), mostly high grass, we used a higher camera altitude to decrease obscured areas. Another possibility is to direct the camera towards nadir (see the manual in Supplementary material S1) to image areas -5 to +5 meters from the center of a plot, further decreasing the viewing angles from nadir. We did not have any problems with shrub or brushwood as it was only a couple of decimeters high, and Birch trees did not grow on the mires. We also recommend using a camera height of about 6 meters to decrease obscuration and to increase the mapped area."

**6. Writing**

The sentences and paragraphs are well written, however in a few cases they start weak with; however, for example, to resolve this, etc. An effort should be made to start sentences with the primary subject of the sentence and tighten up some of the language and remove extra words seen throughout the paper. The acknowledgement heading is floating on page 10 line 15 as well.

**Answer:**

Changes made:

The acknowledgement heading has now been moved closer to the acknowledgement text. The start of sentences have also been improved throughout the paper.

We would like to thank both anonymous reviewers for their valuable comments for improving the manuscript.

[revised manuscript text omitted]

5. Post processing can be done in two ways:

a) Using generally available or commercial software for distortion correction (e.g. Adobe Photoshop, or Adobe Lightroom which has ready-made models for most standard cameras, or the Camera calibration application in Matlab to make a custom model scripts in programming languages) and land cover classification (e.g. CAN-EYE, VegMeasure, SamplePoint, or eCognition). Recalculation of images to overhead projection can be programmed in Matlab using the formulas given in this article (Eqs. 1-2) and using two for-loops to transform each pixel in a lens distortion corrected image to ground coordinates in meters (see Fig. 6).

b) Send images and related information to the lead author who can assist. For this option, we ask image senders to agree on making images and post processed results publicly available via a database hosted by the authors. The aim of this is to over time build a public database that can be used as a source for reference data ground truth archive for remote sensing products.

The post processing software can be requested from the lead author (magnus.galfalk@liu.se). Users should hHowever, please be aware that the software is not user friendly at its present stage as it is not commercial and that there is limited time for support.

**Supplementary  S2 –  additional  examples of classification**

[Figure]

**Figure S1: One of our field plots. (A) Image corrected for lens distortion, with a projected 10 x 10 m grid overlaid. (B) Image after recalculation to overhead projection (10 x 10 m).**

[Figure]

**Figure S2: Close-up images in one of our 10 x 10 m field plots (Fig. S1).**

[Figure]

**Figure S3: Additional eExample of a field plot image with classification into the six main surface components (the plot shown in Fig. S1). All panels have an area of 10 x 10 m. (A) Graminoids. (B) Water. (C) Shrubs. (D) Dry moss. (E) Wet moss. (F) Rock.**

[Figure]

**Figure S4: Additional example of plot classification. (A) Image corrected for lens distortion, with a projected 10 x 10 m grid overlaid. (B) Image after recalculation to overhead projection (10 x 10 m). (C) Graminoids. (D) Wet moss. (E) Water. Images C-E have areas of 10 x 10 m. This plot is clear example of vegetation of different color belonging to the same class as the wet moss included both green-yellow and red vegetation (field notes about the close-up images showed this to be the case).**

[Figure]

**Figure S5: Additional example of plot classification. (A)** Image corrected for lens distortion, with a projected 10 x 10 m grid overlaid. **(B)** Image after recalculation to overhead projection (10 x 10 m). **(C)** Graminoids. **(D)** Shrubs. Images C and D have areas of 10 x 10 m.

[Figure]

**Figure S6: Additional example of plot classification. (A) Image corrected for lens distortion, with a projected 10 x 10 m grid overlaid. (B) Image after recalculation to overhead projection (10 x 10 m). (C) Graminoids. (D) Dry moss. (E) Shrubs. Images C-E have areas of 10 x 10 m.**

---

## Author Response (AR2)

**Revision Biogeosciences**

**Editor comments**

**1.** Page 1/line 12: manufacturer information for the GoPro

**Answer:** The company and camera name is GoPro, the detailed camera name is however HERO4 silver. This has now been added to the manuscript.

Changes made:

- (GoPro HERO4 Silver, GoPro Inc.) added in the manuscript (P1L12)

**2.** 1/17: remove 'education'. The technique seems quite involved with all of the geometric corrections. See also 12/8. 'Specific training' is probably better.

**Answer:** We agree

Changes made:

- Changed education to training (P1L17 and P12L8)

**3.** 1/18: non-forested wetlands.

**Answer:** We agree that the method is mainly suitable for studies of non-forested wetlands

Changes made: The line "can be used as reference data for wetland studies from satellites such as Sentinel 1 and 2 (10 m pixel size)" has been changed to: can be used as reference data for studies of non-forested wetlands from satellites such as Sentinel 1 and 2 (10 m pixel size). (P1L18-19)

**4.** 2/6 'bottleneck' notion could perhaps reference this paper by Dr. P. Crill (https://www.nature.com/articles/nclimate3403.pdf).

**Answer:** We agree

Changes made:

- Crill and Thornton (2017) has been added as reference. (P2L6-7 and references P13L22)

Crill, P. M. and Thornton, B. F.: Whither methane in the IPCC process?, Nature Climate Change, 7, 678–680, 2017.

**5.** 2/11 'northern wetlands' is really boreal, subarctic, and arctic non-forested wetlands. (to a lesser degree these habitats are also present at elevation and in the Southern Hemisphere)

**Answer:** We agree

Changes made: We now explain that our use of "Northern wetlands" refers to "non-forested boreal, subarctic, and arctic wetlands" as it is first mentioned in the introduction (P2L11-12).

**6.** 'The long monopod was made from two ordinary extendable monopods taped together' is unnecessary because single monopods can be of different lengths.

**Answer:** This was written as commercial available monopods are difficult to find that are longer than about 1.8 meters, and an easy solution was to tape together two such long monopods to reach > 3 meter (some overlap required).

Changes made:

- We have changed the sentence to "A long modified monopod with a GoPro camera mounted at the end was used for the imaging" (P4L13-14).

**7.** 'The camera battery typically lasts for a few hours after a full charge' depends on the type of battery used and lithium ion should work better in the cold.

**Answer:** This sentence was meant as a comment on how our work was carried out, using the GoPro 4 Silver camera, that lasted a few hours per charge. Other cameras (different battery types, different age of camera etc.) and different weather conditions could give different battery times. A main point was to show that we charged the camera while travelling between sites, which was enough to last the whole day using only one camera.

Changes made:

- We have now changed 'The camera battery typically lasts…' to 'The camera battery lasted…', using past tense to show that we are writing about our measurements and not camera batteries in general (this also fits well with the rest of sentence 'was charged at intervals when not used'). We also added at the end of the sentence "…making it possible to do all the imaging using only one camera". (P4L16-18)

**8.** Please use the multiplication sign instead of dots in the equations. Dots can also mean complex conjugate.

**Answer:** We agree

Changes made:

- We have now changed the dots to multiplication (x) signs

**9.** 9/20 common abbreviation of 'meters.

**Answer: We agree**

Changes made:

- 'meters' changed to 'm' throughout the manuscript

[revised manuscript text omitted]